# Effect of Agricultural and Urban Infrastructure on River Basin Delineation and Surface Water Availability: Case of the Culiacan River Basin

**Sergio A. Rentería-Guevara** [1,2], **Jesús G. Rangel-Peraza** [1,*], **Abraham E. Rodríguez-Mata** [3], **Leonel E. Amábilis-Sosa** [3], **Antonio J. Sanhouse-García** [1] **and Perla Marysol Uriarte-Aceves** [1]

[1]   Doctorado en Ciencias de la Ingeniería, Tecnológico Nacional de México/Instituto Tecnológico de Culiacán, Juan de Dios Bátiz 310, Col. Guadalupe, CP 80220 Culiacán Rosales, Sinaloa, Mexico
[2]   Facultad de Ingeniería, Universidad Autónoma de Sinaloa, Calzada de las Américas Nte s/n, Cd. Universitaria, CP 80013 Culiacán Rosales, Sinaloa, Mexico
[3]   CONACYT-Instituto Tecnológico de Culiacán, Juan de Dios Bátiz 310, Col. Guadalupe, CP 80220 Culiacán Rosales, Sinaloa, Mexico
*   Correspondence: jesus.rangel@itculiacan.edu.mx

**Abstract:** River basin delineation can be inappropriate to determine surface water availability in a country, even if it is established by its water authority. This is because the effect of agricultural and urban infrastructure in runoff direction is ignored, and the anthropogenic changes in hydrography and topography features distort the runoff. This situation is really important because water rights are granted based on volumes that are not physically accessible. The existence of this problem is demonstrated through a case of study: the Culiacan River Basin in Mexico. To overcome such a situation, this study poses criteria to revise official river basin configurations and to delineate new river basins based on digital elevation models, vector files of agricultural infrastructure, and extensive field verification. Significant differences were noticed in surface water availability calculated under distinct river basin delineations.

**Keywords:** river basin; delineation; water-balance; drainage divide

## 1. Introduction

### 1.1. Official River Basin Delineation

A river basin is a territory delimited by nature itself, essentially by the limits of surface runoff zones that converge towards a single watercourse [1]. Despite the natural character of a drainage basin, a territory can be divided into different hydrological configurations, as is observed in Mexico [2]. Therefore, scientists and decision-makers must deal with the absence of consensual river basin delineation within a country.

To overcome this problem, some countries divide their territory by an official delineation of drainage basins, which intends to be a reference for water distribution. This official connotation of a river basin is necessary because it is used as an official tool to manage water resources, such as the basins of the National Water Bank (NWB) in Chile [3]. In Mexico, the water authority (Comision Nacional del Agua, CONAGUA) established an official delineation of river basins. Its use is obligatory to evaluate official surface water availability, which is the basis to confer or deny water rights. However, performing a hydrological division for this purpose implies coping with some situations that cannot be overlooked.

*1.2. Possible Situations for a River Basin Modification*

### 1.2.1. Topography and Hydrography

The natural flow pattern should be the basis on which to perform a drainage basin delineation. Therefore, official basin delineation must be consistent with topography and hydrography. In addition, the territory must accomplish the condition of draining through its main watercourse towards a single outlet [4]. This outlet is a control point from which to calculate the surface water balance. The set of magnitudes of the input variables and the results are valid up to that point.

In Mexico, a drainage divide based solely on these natural criteria was broadly published by a geography government agency [5]. However, official delineation generated by the water authority [6] does not coincide with that natural delineation as described below. Besides, it is a common practice for the water authority to report a water balance in a drainage unit where several outlets are identified.

### 1.2.2. Lack of Hydrometric Data

When a water balance is performed in a river basin, it should be considered that runoff can be generated or dissipated within the basin or it can cross the basin boundary. To quantify this component of the water balance, gauging infrastructure is required such as hydrometric stations or flow meter devices located in the reservoirs. In drainage basins with insufficient data of water flows or without the presence of gauging stations, surface water availability determination is carried out by estimating runoff through indirect methods. These methods are based on precipitation records and information about land use and type of soil, among other topographic characteristics. However, such estimation reduces the precision and reliability of the water balance [7]. Therefore, the use of hydrometric data is preferable to reliance on precipitation information to evaluate surface water availability.

Thus, the scarcity or absence of hydrometric data in some regions is a fact that needs to be addressed. In Colombia, one of the major difficulties in estimating the water availability is the lack of information because there are few climatologic and meteorological stations [8,9]. In Mexico, the use of official data from hydrometric stations for different purposes, such as hydrological forecasting or water balance calculations, could face several problems: irregular distribution of the hydrometric stations throughout the territory, different observation periods, a drastic decrease in the number of stations over the last 30 years, and even gaps in data are observed [10,11]. Therefore, in these regions, it is a common practice to delineate watersheds in locations where hydrometric stations or reservoirs are found to take advantage of the hydrometric information of gauging stations [12].

### 1.2.3. Presence of Hydraulic Infrastructure in Drainage Basin Delineation

The presence of hydraulic infrastructure to supply or to drain water from zones having agricultural [13] or urban [14] development tends to modify their natural water discharge pattern. Conveyance channels supply surface water to crops and drainage channels (agricultural drains) and remove excesses water from crops, while storm drainage directs flows away from urbanized areas. In zones with intense agricultural operations or urban infrastructure, these channels constitute an artificial channel network that interacts with natural watercourses. Such an interaction defines the real flow pattern of these areas.

*1.3. Investigation Objectives and Novelty*

River basin delineation is critical to carry out a water balance. In this respect, it is a common practice that water authorities or governments establish official river basin delineations and no critical analysis of the official information is presented. This study revises official drainage basin delineation of the Culiacan River Basin in Mexico. This revision is carried out to test its suitability to properly perform a water balance. One objective of this paper is to demonstrate that both official and natural basin delineations are not appropriate to perform a water balance in the study area due to the presence of agricultural infrastructure, and consequently, surface water availability is miscalculated. The other

objective is to propose proper basin delineation aimed at carrying out a surface water balance closer to physical reality and useful to determine the official availability of water in this region.

## 2. Materials and Methods

### 2.1. Study Area

As the case study, the Culiacan River Basin is located between coordinates 105°41′ and 108°4′ W and 24°24′ and 26°24′ N. This basin's hydrography is comprised of three main rivers: Humaya and Tamazula, whose confluence forms the Culiacan River in the city of the same name (Figure 1). Two water bodies are identified in this basin: Adolfo Lopez Mateos and Sanalona reservoirs. They are located in the watercourses of the Humaya and Tamazula rivers, respectively, which comprise the main water supply for irrigated agriculture. The Culiacan River Basin extends up to the Culiacan River to the sea through a cove named Ensenada Pabellones. In Figure 1, the red dots indicate the two outlets of the Culiacan River Basin.

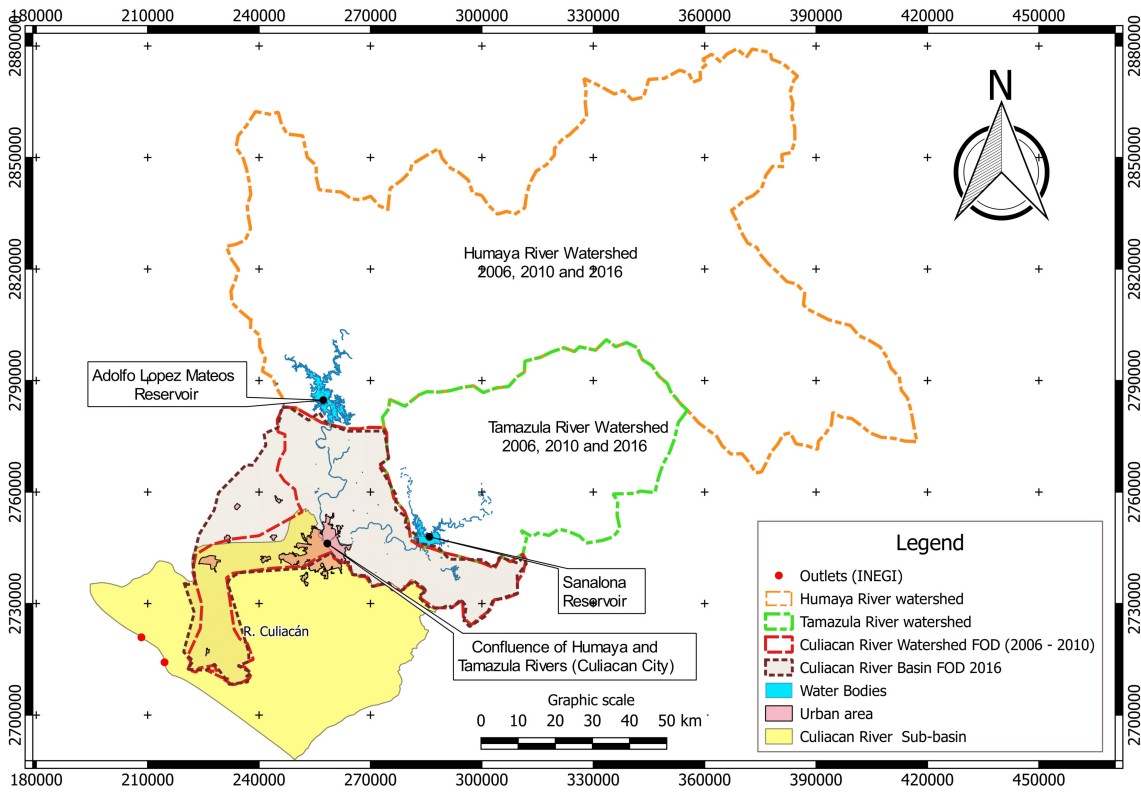

**Figure 1.** Culiacan River Watershed according to the shape-files of INEGI (2017a) and the coordinates of vertexes published in CONAGUA (2016a).

Despite the study area defined by INEGI (Instituto Nacional de Estadística e Informática) being based on natural hydrologic division, the National Water Commission [6] established in 2016 a delineation based on three drainage basins (Humaya River, Tamazula River, and Culiacan River basins) that does not correspond to the INEGI delineation (Figure 1). In CONAGUA's delineation, the Humaya River Basin does not coincide with that proposed by INEGI. The official delineation (CONAGUA) includes part of the Humaya River Basin up to Adolfo Lopez Mateos Reservoir. The same situation is observed for the Tamazula River Basin, where the official delineation partially encompasses this basin up to Sanalona Reservoir. Then, based on the CONAGUA delineation, the Culiacan River Basin initiates at the Adolfo Lopez Mateos and Sanalona reservoirs and ends in their outlets to the sea.

In Figure 1, it should be also noticed that CONAGUA previously established a watershed division of the same area [15,16]. The official drainage basin delineation of the years 2006, 2010, and 2016

matches the Humaya and Tamazula basins, but it does not correspond with the Culiacan River Basin (Figure 1). The difference is substantial in the lower part of this basin. A wide area located to the northwest of the lower part was added in the basin delineation of 2016. The official publications do not explain the reason for this change.

## 2.2. Methods to Determine Official Availability of Surface Water

In Mexico, official norm NOM-011-CONAGUA-2015 [17] must be used to determine the official annual mean availability of surface water. The surface water balance in the basin of interest was based on the hydrological interpretation of the continuity equation [18].

$$\Delta V_t = [Q_P(t) + Q_R(t) + Q_{ALM}(t) + Q_{SL}(t) + Q_{SN}(t)]_{inlets} - [Q_{EX}(t) + Q_{EV}(t) + Q_{OUT}(t)]_{outlets} \quad (1)$$

In Equation (1), $\Delta V_t$ refers to the annual storage variation of the Culiacan River; $Q_P(t)$ is the annual average precipitation volume for the year $t$; $Q_R(t)$ is the annual average natural runoff volume for the year $t$; $Q_{ALM}(t)$ is the annual average inlet volume from Adolfo Lopez Mateos reservoir for the year $t$; $Q_{SL}(t)$ is the annual average inlet volume from the San Lorenzo irrigation channel for the year $t$; $Q_{SN}(t)$ is the annual average inlet volume from Sanalona reservoir for the year $t$; $Q_{EX}(t)$ is the annual average water extraction volume for the year $t$; $Q_{EV}(t)$ is the annual average evaporation volume for the year $t$; and $Q_{OUT}(t)$ is the annual average outlet volume of Culiacan River for the year $t$. In this study, a water balance was carried out for the period ($t$) of January–December 2016.

This water balance was supported by a rain-runoff relation, where runoff is a fraction of annual precipitation volume, and it is also calculated based on the basin delineation. International literature reports diverse methods to estimate this relationship [7,19]. Therefore, the river basin boundaries are critical to identify the terms of the continuity equation and to define their magnitude. If the catchment area is augmented, terms like runoff increase. The evaporation and evapotranspiration terms depend on the presence of water bodies and vegetation within the limits of the basin.

Generally, each term is identified and quantified with reference to watershed delineation. If a gauging station is not present in a watershed border, which is a common case in river basins flowing directly to the sea, the estimated annual runoff volume depends on the calculated basin area, which in turn is determined by its delineation. This is the case of the drainage basin under study. According to CONAGUA, there are gauging devices in the two reservoirs located in the upstream zone of the Culiacan River Basin, but there is no flow measurement downstream. This means that annual natural runoff volume of the basin should be calculated by the following mathematical expression [20]:

$$Q_R(t) = H_p * A * \alpha \quad (2)$$

where $H_p$ is the annual average precipitation on the basin; $A$ is the watershed area and $\alpha$ is the runoff coefficient. According to NOM-011-CONAGUA-2015, runoff is estimated using the Soil Conservation Service (SCS) method. Annual natural runoff volume of the watershed is directly proportional to the watershed area and precipitation. This term is calculated by correcting this value by means of a runoff coefficient, considering the type of soil and the land use. This information was obtained from Sanhouse-García et al. [21] in the study area. Due to surface water availability being calculated for a one-year period, a constant runoff coefficient was used. However, this coefficient should be updated every year for larger study periods.

## 2.3. Databases, Software, and Information Process

### 2.3.1. Databases and Software

The dataset used in this study was obtained from the Mexican government repositories. The information included: hydrography, localization of watershed outlets, digital elevation models (DEM), watershed delineations, localization of gauging stations, and the channel network of the

irrigated area. Vector files in shape format (*.shp) were retrieved from the website of the National Institute of Statistics and Geography [5]. They corresponded to Hydrographic Network Scale 1:50,000 of Hydrological Region 10 Sinaloa, watershed outlets, and natural river basin delineation. DEM raster files in "Band Interleaved by Line" format (*.bil) were also obtained from the website of INEGI [22]. The official delineation of hydrological basins in Mexico was based on vertex coordinates published in text Acrobat format (*.pdf) in the Federation Official Journal [6]. Hydrometric station coordinates were consulted in the Surface Water National Data Bank [23]. In addition, vector files of the channel network of the study zone were obtained from the Irrigation District No. 10 Office [24].

Dataset processing to delineate river basins was based on integrating two separate information systems that share the same user interface: Quantum GIS [25] Las Palmas 2.18.3 and GRASS [26]. Digital image processing for watershed delineation was performed in two phases: basic cartographic information processing and application of the algorithm r.watershed.

### 2.3.2. Basic Cartography

This phase involved retrieving and editing the basic information mentioned above. Georeferencing of the basic cartography was carried out using the WGS84 reference system (World Geodetic System 1984) and UTM projection (Universal Transverse Mercator), Zone 13. Official watershed divides were delineated based on vertex coordinates published in pdf format converted to Excel files. Then, they were saved in comma delimitated text format and added to the GIS canvas using function "Add comma delimitated text layer". Terrain elevation curves and labels were generated from elevation data using GIS capabilities.

### 2.3.3. Hydrological Analysis Algorithm

The r.watershed algorithm was executed from graphic interface GRASS through the "Process Toolbox". By executing this algorithm, a set of outputs was generated: (1) pre-processing of DEM (depression filling), (2) flow direction, (3) flow accumulation, (4) localization of watercourses, and (5) river basin delineation. Nevertheless, before executing this algorithm, it is necessary to load a DEM.

The procedure to delineate and to analyze river basins considered the raster model as an input. Contour lines were generated by using the specialized module r.surf.contour based on linear interpolation. The elevation of a point on a contour map was determined by interpolating the value of the two nearest contour lines (uphill and downhill).

River basin delineation was achieved drawing a polygon that surrounded the catchment. This was calculated based on the total size of the input raster from which half-basin outputs were generated. In this study, the size of the input raster was used to calculate river basins with the rewire outlet algorithm. The result was an output raster where each pixel had an assigned value that indicated the catchment to which it corresponded. Catchments were polygonized, and a conversion to a vector file (shapefile) was then carried out.

### 2.4. Proposed River Basin Delineation

Since the Mexican Government delineations do not reflect the influence of urban or agricultural infrastructure, a different watershed divide was proposed to represent real flow patterns. First, DEMs were processed with a GIS to generate the sub-basin divide and the level curves of the study area. This information was critical because watershed delineation was then drawn on screen by hand. This drainage divide was carried out taking into account the topography and hydrography features of the study area and the presence of urban and agricultural hydraulic infrastructure. In particular, irrigation and drainage channel networks were used as the basis for mapping drainage divides. Besides, the satellite images and field inspection provided information about additional infrastructure, such as urban storm drainage systems and shrimp farms.

The starting point to map the drainage divide was the river basin outlet. In this zone, GIS failed at mapping drainage divides because the terrain was flat. Polygons were drawn by hand based on

the highest elevations observed in level curves. In this area, satellite images and field inspection were used to identify water flow patterns induced by the presence of shrimp farms (Figure 2).

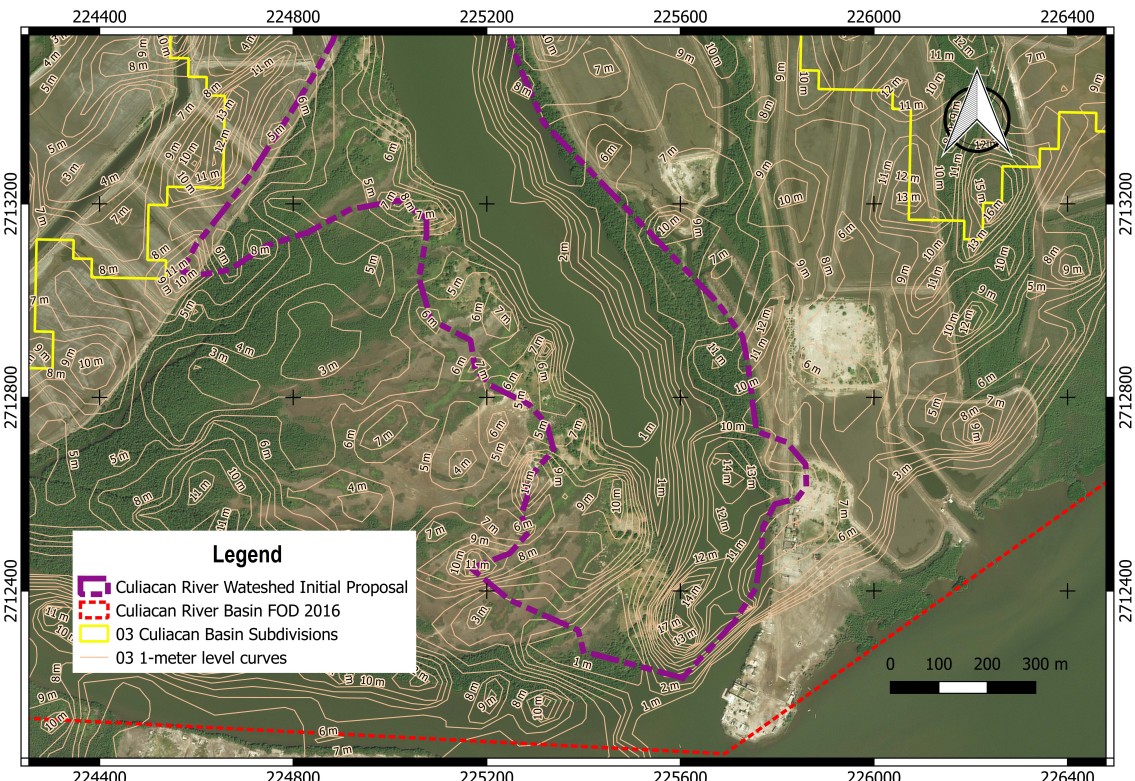

**Figure 2.** Use of level curves and watershed divides generated by GIS to propose an initial watershed divide.

Upstream, where agricultural infrastructure becomes relevant, the drainage channel influence was considered. These channels were built in swathes of territory of relatively high elevations with respect to areas along the rivers. Therefore, they collect water that originally flowed to the main watershed channel (river). This situation defined the proposed watershed divide, which is physically pertinent in the study area. The initial drainage divide was proposed by a boundary line that followed the agricultural drainage channels' borders. This situation implied that the natural (topographic) catchment area was suppressed due to the presence of this infrastructure. This situation is explained in Case a shown in Figure 3.

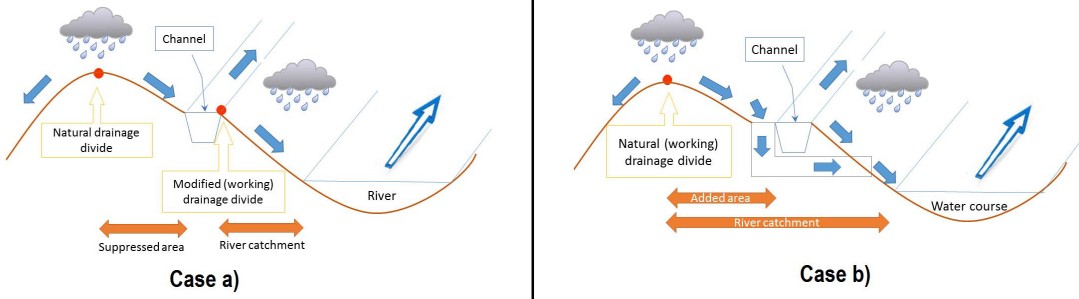

**Figure 3.** Criteria used to add or suppress catchment area in the proposed division.

In contrast, the field verification identified that there were crossing structures that conveyed streamflow to the opposite side of channels. In such cases, the natural catchment area was restored to the proposed watershed divide, as Case b of Figure 3 explains.

## 3. Results and Discussion

### 3.1. Culiacan River Basin as a Drainage Unit

According to the INEGI delineation, the study area has two outlets, marked with red circles located in the downstream part of the Culiacan River Basin (Figure 4). In terms of the water balance, this area consists of at least two different drainage entities requiring separate calculations.

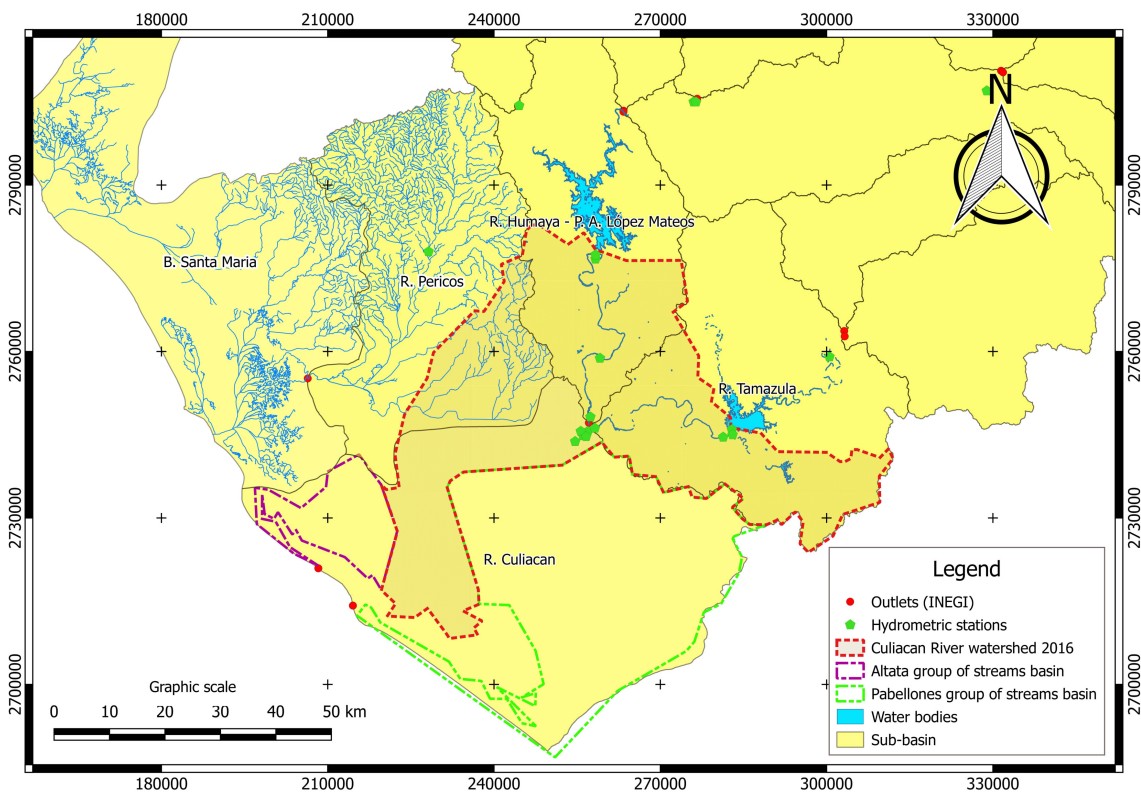

**Figure 4.** Culiacan River Watershed and Pericos River Basin according to INEGI (2017a); "Pabellones Group of Streams" and "Altata Group of Streams" basins according to CONAGUA (2016a); hydrometric stations and drainage sites obtained from CONAGUA (2017a).

However, CONAGUA delineation of the Culiacan River Basin in 2016 included an area identified as Pericos River Basin that does not drain to the Culiacan River. This area drains independently towards the sea, as shown in Figure 4. This means that Pericos River Basin physically does not correspond to the Culiacan River Basin. Consequently, this delineation is not appropriate to perform a water balance having a basin as a drainage unit. The delineation inconsistencies of Culiacan River Basin were depicted by comparing this delineation to publications issued in 2006 and 2010. These versions approximate to the hydrography of the area (Figures 1 and 4). Despite CONAGUA delineations in 2006 and 2010 excluding the Pericos River Basin, the official basin differs noticeably from the natural watershed in the lower part because CONAGUA delineation narrows towards the Culiacan River. The additional areas correspond to the Group of Streams Pabellones and Group of Streams Altata basins (Figure 4). Thus, their runoff drains towards the sea independently of the Culiacan River. Therefore, in the coast limits, there are at least three drainage sites (and not two as INEGI establishes), which correspond to the same number of basins (Culiacan River, Group of Streams Pabellones, and Group of Streams Altata). Each of these basins requires separate balances to calculate their surface water availability.

### 3.2. Consistency with Topography

A sub-basin size delineation of the study area was compared with the official delineation of the Humaya, Tamazula, and Culiacan river basins (Figure 5). In this figure, discrepancies are identified in the lower part of the Culiacan River watershed. Sinuous lines represent the highest elevations in the territory (drainage divide). These lines should match the official delineation established by CONAGUA. However, official basin delineation does not match in the downstream of the Culiacan River Basin. Sinuous lines are not followed by CONAGUA delineation. This situation indicates that official watershed delineation is not appropriate because it does not constitute a topographic boundary and it does not physically divide the runoff between adjacent basins.

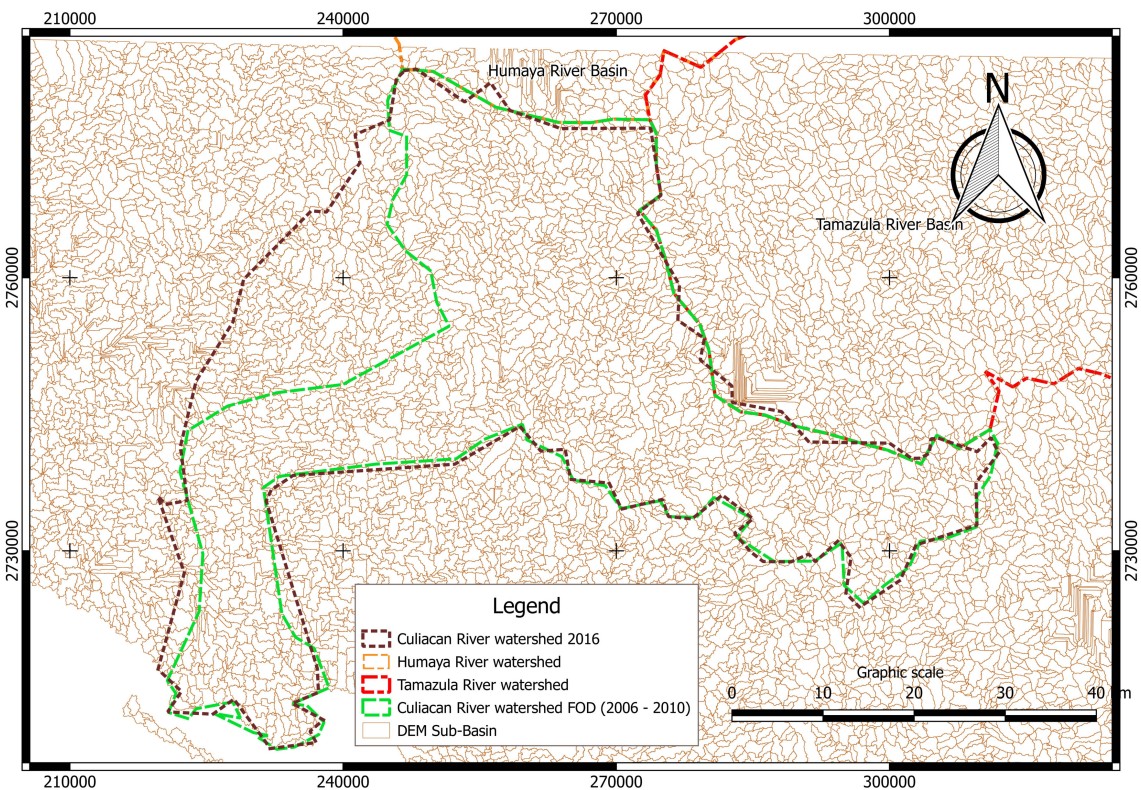

**Figure 5.** Comparison between drainage divides generated with a GIS based on DEMs obtained from INEGI (2017b) and the official watershed delineation published by CONAGUA (2016a).

### 3.3. Presence of Flow Gauging Devices

Hydrometric stations located in the Adolfo Lopez Mateos and Sanalona reservoirs generate the latest and more complete information of the study area according to CONAGUA [22]. However, these reservoirs are located within the Humaya and Tamazula river basins, respectively. To take advantage of the presence of the hydrometric infrastructure, water authorities in Mexico decided to delineate the boundaries of both basins in the dam walls, where the gauging stations are located. Thus, such information is used to estimate surface water availability in the entire Culiacan River Basin (Figure 6).

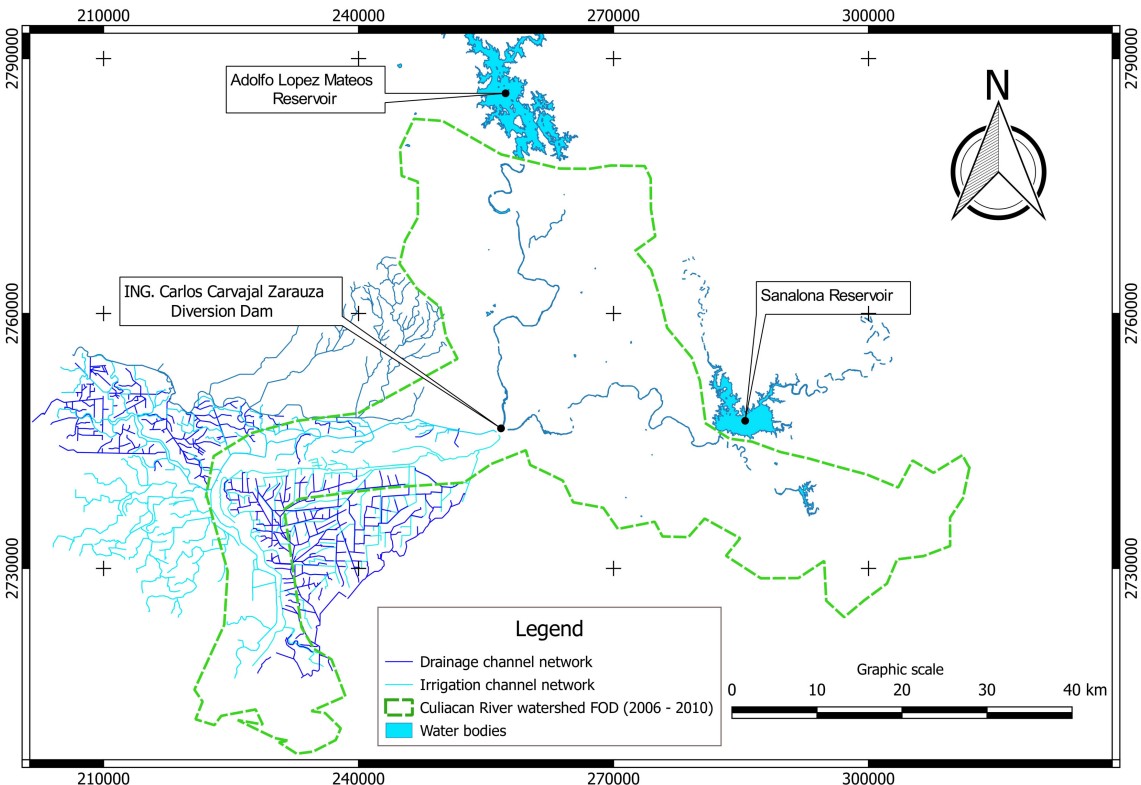

**Figure 6.** Culiacan River watersheds published by CONAGUA (2016a) in 2006 and 2010; hydrography of the study area (INEGI, 2017a) and drainage channel networks of Irrigation District 010 Culiacan Humaya (CONAGUA, 2017b).

### 3.4. Influence of Agricultural Infrastructure

Agricultural infrastructure affects the drainage divide delineation of the Culiacan River Basin (Figure 6). As stated previously in Section 2.4, channels must be considered to delineate a drainage divide. This is because the presence of these channels diverts water from its natural course. Figure 7 shows the case of a stream that originally drained to the Culiacan River (arrows indicate the flow direction). However, its trajectory was interrupted by an irrigation channel (Point 1). On the left side of the conveyance conduction channel, a natural drainage channel was identified, but originally, this watercourse was the continuation of the stream. This situation is common in the irrigated areas located in the lower part of the Culiacan River Basin, where agricultural infrastructure modifies the original direction of runoff. Consequently, drainage divide delineation of this area should be modified under these conditions.

The main proposed modifications of the natural drainage divide of the Culiacan River Basin are due to the presence of two irrigation channels. They begin from a diversion dam named Ing.Carlos Carvajal Zarazua and continue at a short distance along both sides of the Culiacan River, almost until its outlet to the sea. Because these channels divert runoff inputs to the river, these channels are the basis to delimitate the water parting of Culiacan River watershed, which is physically pertinent in this area.

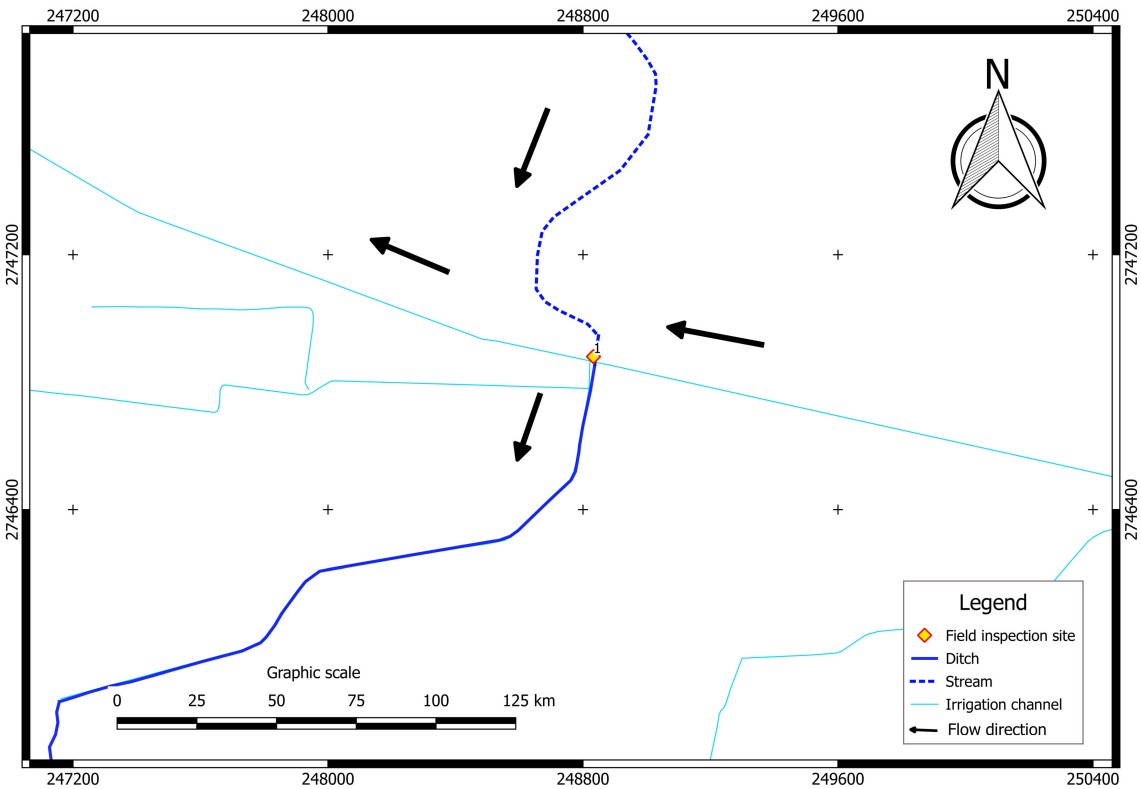

**Figure 7.** A stream drains to a channel in the study area.

### 3.5. Proposed Delineation of the Culiacan River Basin

Based on the above basin delineation analysis, it is evident that the CONAGUA [6] and INEGI [5] delineations are not appropriate for a water balance for the Culiacan River Basin. A new delineation is convenient, based on the study area characteristics. This new basin delineation considers the following criteria: (1) maintaining the condition of a single outlet for basin runoff and keeping the natural basin delineation regarding topography, (2) considering most of the accessible gauging stations, and (3) considering hydraulic infrastructure as an operational physical basin boundary.

For this purpose, a single outlet point of the Culiacan River Basin was defined at its downstream limit. The Culiacan River Basin delineation generated with GIS gave the natural basis of the delineation to be proposed. The dam walls of Adolfo Lopez Mateos and Sanalona reservoir were used to delimitate the upstream border of the basin because the most complete hydrometric information of the area was generated at these locations. Irrigation and drainage channel networks, as well as storm drainage conduits were considered for the final basin proposal in the rural and urban areas of the Culiacan River Basin. Taking into consideration these conditions and using the GIS, an initial drainage basin delineation proposal was generated by hand (Figure 8).

Due to limitations in the resolution of satellite images, 12 sites were inspected in the field to confirm if natural or artificial watercourses crossed the irrigation channels or they only drained to them (Table 1).

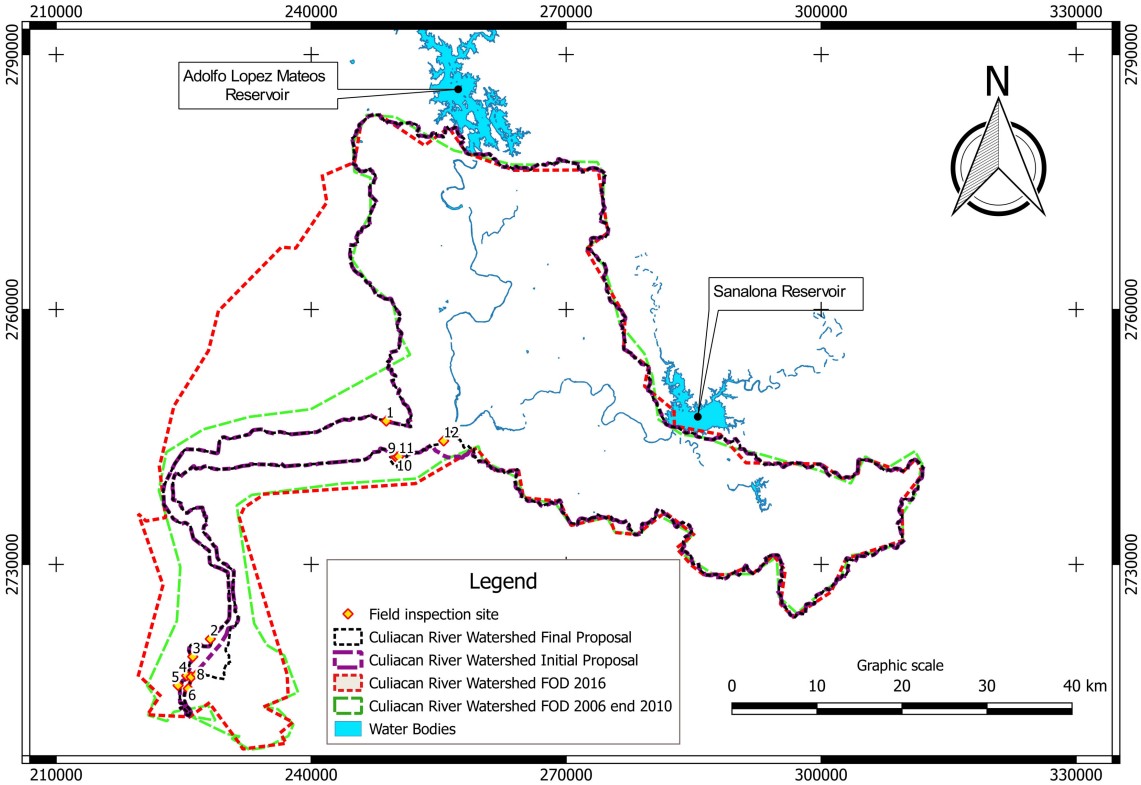

**Figure 8.** New proposal of the Culiacan River Watershed with some verification field points.

**Table 1.** Location of the verification sites to identify possible modifications in the watershed divide proposal.

| ID | Longitude | Latitude | ID | Longitude | Latitude | ID | Longitude | Latitude |
|----|-----------|----------|----|-----------|----------|----|-----------|----------|
| 1 | 248,832.8936 | 12,746,880.34 | 5 | 224,344.4859 | 12,715,766.51 | 9 | 249,924.9679 | 12,742,693.26 |
| 2 | 228,135.6428 | 12,721,232.2 | 6 | 225,541.8782 | 12,715,385.57 | 10 | 250,048.2476 | 12,742,639.38 |
| 3 | 226,070.8391 | 12,719,149.27 | 7 | 225,891.3327 | 12,716,760.68 | 11 | 250,302.4689 | 12,742,703.47 |
| 4 | 225,354.337 | 12,716,909.94 | 8 | 226,540.4675 | 12,717,061.18 | 12 | 255,585.3322 | 12,744,541.77 |

This validation could only be performed through on-site inspection to include or exclude the runoff generation area. Only four sites motivated modifications in the original basin proposal (Table 2).

**Table 2.** Modified sites from the initial proposal.

| ID | Longitude | Latitude | Local Reference | Observations |
|----|-----------|----------|-----------------|--------------|
| 7 | 225,891.3327 | 12,716,760.68 | Intersection between a ditch and a road. | Field inspection confirmed that the ditch crosses underneath the road. Thus, the watershed surface area was extended to include the catchment area of the ditch (Case b of Figure 3). |
| 8 | 226,540.4675 | 12,717,061.18 | El Castillo | The shrimp farm ditch crosses the road flowing to the Culiacan River. The drainage watershed was extended to include the catchment area of the ditch (Case b of Figure 3). |
| 9 | 249,924.9679 | 12,742,693.26 | Bachigualato Channel Bridge | Storm flows coming from airport zone and the surface runoff coming from the Bachigualato area have a confluence. This runoff confluence continues to a culvert under a railroad that continues through an open lined channel under a bridge canal. The drainage divide was extended to include the corresponding catchment (Case b of Figure 3). |
| 12 | 255,585.3322 | 12,744,541.77 | Federalismo Avenue | Runoff drains to the Oriental Main Channel. Thus, the drainage divide was reduced to exclude this surface catchment area (Case a of Figure 3) |

These modifications were minimal considering the total surface of the initial river basin proposal, but significant as a procedural basis for future analysis of drainage divides. A comparison between the initial and final watershed delineation proposals is shown in Figure 8.

Figure 9 shows one of the sites (Site 9) that required modifications of the initial basin delineation proposal. The field recognition revealed that runoff coming from Bachigualato and Culiacan city airport areas drains to the Culiacan River due to agricultural and urban infrastructure. Figure 9a shows that runoff goes through a culvert underneath a railroad. Downstream, water flow is conveyed underneath a bridge channel through a lined channel (Figure 9b,c) and then directed to the Culiacan River through an unlined channel (Figure 9d).

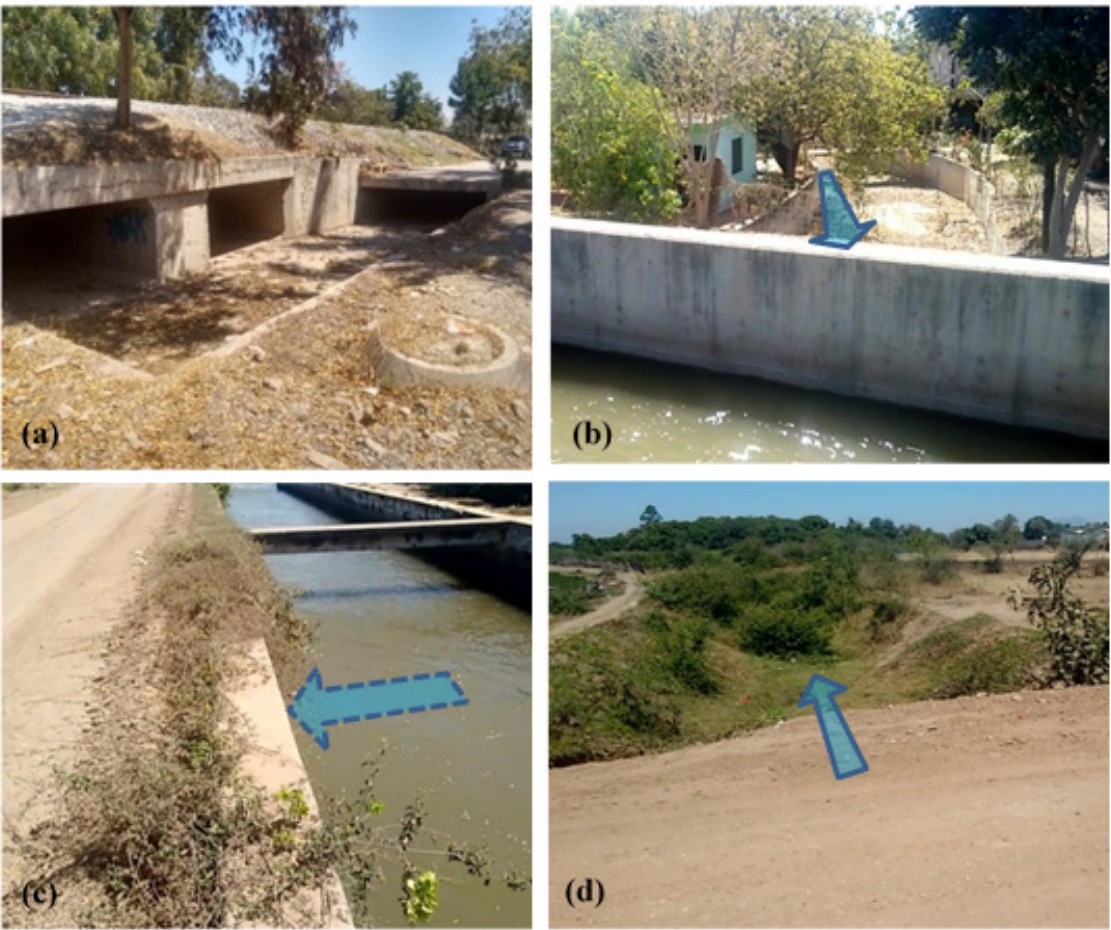

**Figure 9.** Evidence of changes in the drainage pattern due to agricultural and urban infrastructure. (**a**) Culvert drains runoff under a railroad. (**b**) Concrete-lined open ditch passes under a channel bridge. (**c**) Runoff drains to lateral channel. (**d**) Unlined ditch passing under a channel bridge.

Once the field verification confirmed that water flow crosses to the opposite side of the irrigation channel, the corresponding catchment area was added to the initial proposal. This additional area was defined using GIS capabilities. The watercourses and the sub-basin drainage divides were the basis for hand-drawing on screen the new boundaries (Figure 10).

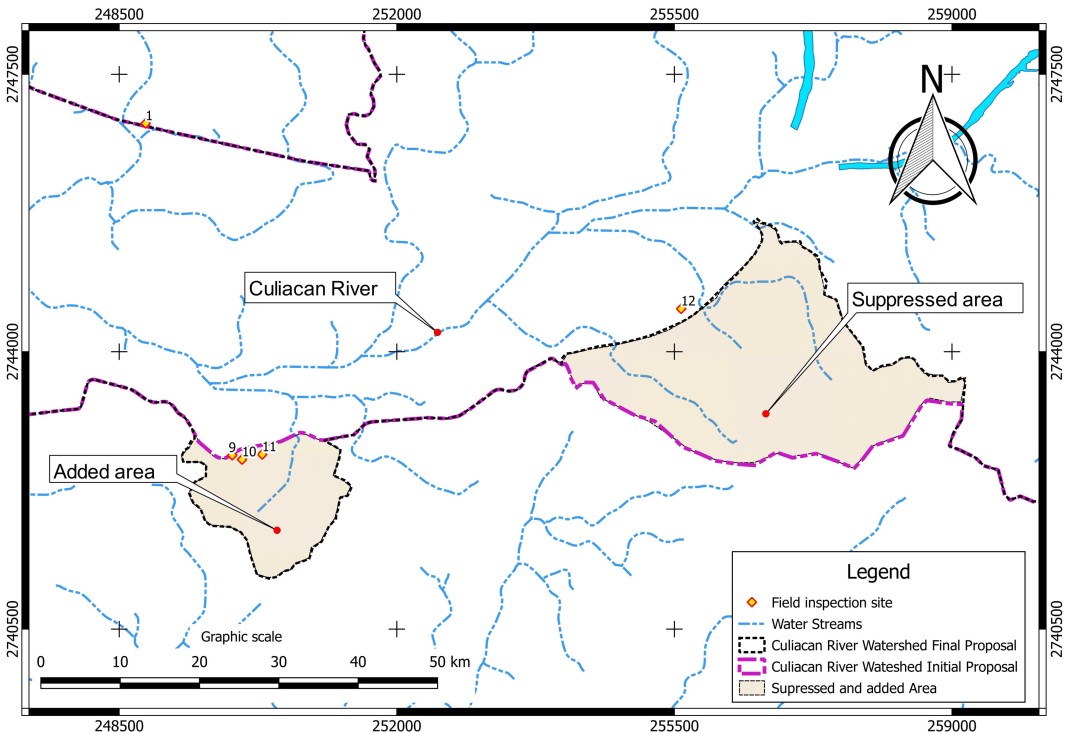

**Figure 10.** Culiacan River drainage divide, including the urban and agricultural infrastructure in the study area.

### *3.6. Effect of River Basin Delineation in the Estimation of Surface Water Availability*

The catchment areas for the different basin delineations are shown in Figure 8. The basin area proposed in this study was 1688.2 km$^2$, while catchment areas of 2625.8 km$^2$ and 2145.5 km$^2$ corresponded to the basins proposed by the National Water Commission in 2016 and 2010, respectively. According to official delineation of 2016, the magnitude of this basin natural runoff was 466.13 Mm$^3$ per year [27], which was reduced to 266.7 Mm$^3$ per year when using the final proposed delimitation (Table 3). This means that the annual natural runoff volume was overestimated by up to 27.1% with respect to the basin official delineation given in 2010 and by up to 55.5% with respect to the current watershed delineation (2016).

**Table 3.** Annual mean surface water availability of the Culiacan River (in Mm$^3$/year).

| Source | $Q_P(t)$ | $Q_R(t)$ | $Q_{ALM}(t)$ | $Q_{SN}(t)$ | $Q_{SL}(t)$ | $Q_{EX}(t)$ | $Q_{EV}(t)$ | $Q_{OUT}(t)$ |
|---|---|---|---|---|---|---|---|---|
| CONAGUA (2016b) | 4.71 | 466.13 | 1430.65 | 702.5 | 312.74 | 2632.8 | 14.68 | 269.25 |
| Final proposal (this study) | 4.71 | 266.7 | 1430.65 | 702.5 | 312.74 | 2632.8 | 14.68 | 102.92 |

In this study, annual storage variation was not significant ($\Delta V = 0$) because the Adolfo Lopez Mateos and Sanalona reservoirs are located outside of the Culiacan River Basin, and the storage capacity of the river course and the diversion dam within the basin was not significant. Thus, the following equation can be deduced:

$$Q_{OUT}(t) = Q_P(t) + Q_R(t) + Q_{ALM}(t) + Q_{SN}(t) + Q_{SL}(t) - Q_{EX}(t) - Q_{EV}(t) \tag{3}$$

This form of the continuity equation suggests that the annual surface water availability of the basin can be calculated in terms of $Q_{out}$. According to official information, surface water availability is 269.25 Mm$^3$ per year. When the new delineation was applied, the result of the surface water balance

in the Culiacan River Basin was an annual mean surface water availability of 102.92 Mm$^3$ per year. This result was 166.33 Mm$^3$ per year less than that officially published in 2016.

By law, the water authority has the obligation to confer rights to use all the available water according to water balances published in the Federation Official Diary. This situation implies that if authorities grant water rights based on the official available volume, an over exploitation of water in administrative terms occurs. The river basin under study demands large volumes of water for agriculture activities. A miscalculation in water balance can generate severe conflicts among water users because the authority is obligated to grant rights to use quantities of water that physically are not available.

## 4. Conclusions

A river basin delineated by water authorities or geographic agencies cannot be supposed as a orrect geographical frame to calculate surface water availability. These delineations can have serious inconsistencies, such as: including a significant area that drains to the sea independently of its main watercourse, considering its natural drainage divide as an operational physical boundary where important irrigation channels function as real drainage divide and mismatching drainage divides generated with DEMs published by the authority itself.

An alternative basin delineation was proposed that overcame these inconsistencies. Considering this last delineation for the case of study, surface water availability was 113.0 Mm$^3$/year (millions of cubic meters per year), which was 166.0 Mm$^3$/year less than the 279.0 Mm$^3$/year published in the Federation Official Diary. This means that official water availability in this case was overestimated, and water rights can be granted to use annual volumes that in practice are not accessible.

The present study could also explain the water deficits that historically have been found in several hydrological basins in Mexico and that have originated different conflicts for the water users. Likewise, this work provides the basis for improving water balances carried out in regions where a strong pressure for water resources is present due to the lack of gauging stations or the presence of agricultural infrastructure.

**Author Contributions:** Formal analysis, A.J.S.-G.; Investigation, A.E.R.-M.; Methodology, L.E.A.-S.; Software, P.M.U.-A.; Writing—original draft, S.A.R.-G.; Writing—review & editing, J.G.R.-P.

**Funding:** This research was funded by CONACYT grant number Cátedras CONACYT 2014 Ref. 2572.

**Acknowledgments:** The authors would like to thank Francisco Armando Chávez Durán, commissioned by CONAGUA, for providing support in field research and data for analysis.

**Conflicts of Interest:** The authors declare no conflict of interest.

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
