# Peer review of "Effect of Agricultural and Urban Infrastructure on River Basin Delineation and Surface Water Availability: Case of the Culiacan River Basin"

_hydrology, doi:10.3390/hydrology6030058_

Round 1

Reviewer 1 Report

This paper ma amkes an important point about the unreliability of some water allocation schemes based on unrealistic estimates of water availability because human diversions of water an alterations to drainage networks are not taken into account.  Some of the wording is a little confusing, particularly what is meant by "drainage points".  Are these flow monitoring sites (gauging stations) or are they just confluences where one river joins another, or the sea, or enters a reservoir?  It would be advisable to check the paper carefully to make sure that the meaning of the terms you use is absolutely clear.

The point about field checking of the actual channels is very important.  Perhaps you could clarify that the GIS work was based solely on the topography and that a separate mapping exercise was necessary to take into account all the constructed irrigation and urban drainage channels.  Are the any sewers and water treatment works that cause transfers of water from one basin to another (i.e. release of treated waste water into a river that is different from that where the water supply comes from).  None of the maps shows the extent of the urban areas or the actual irrigation areas.

I have annotated the attached pdf, but there are many other places where the language should be carefully checked.

Reviewer 2 Report

General comments

            It is possible that some watershed delineations from authority sources are incorrect. This manuscript deals with such case in Culiacan River Basin, Mexico, and corrects it with considering agricultural and urban infrastructure within the watershed. The results were validated by site inspection. Based on the modified watershed boundaries, the surface water availability can be properly calculated. The proposed delineation method comprises of two steps: delineating using a GIS approach with DEM and post-processing with knowledge on the local drainage system such as the topology of channels. The proposed method is actually not novel and in practice we usually take similar methods. But this work will be useful for local water resources management. The writing has space to improve (such as too much on the GIS delineation and not clear on the proposed method). The figures are in low quality. The estimation of surface water availability is not scientifically sound. The current manuscript looks more like a technical report rather than a research paper. I recommend a major revision.

Some specific comments

1. All the figures are in low quality and need to be replotted at a higher resolution. Legends are required for all the maps.

2. In section 2.3, too much on the GIS delineation which is a common skill. Figure 2 is not necessary. The steps or guidelines for the proposed river basin delineation in Section 2.4 are expected. More details should be provided in this section.

3. How to use the satellite images for delineating the basin boundaries? It is mentioned many times in the manuscript but not used in the case study. It requires a more specific explanation.

4. Figure 9. It might be OK in this specific case but it may depend. How do you determine the added or suppressed boundaries? I hope there will be some guidelines or principles for handling those modifications covering most cases and it will add good values for sure to this manuscript.

5. Line 123, the variable A should be watershed area. A constant runoff coefficient may be not applicable for every year in a large basin so the estimation in section 3.6 is not reliable. In other words, the incorrect watershed boundaries do affect the estimation of surface water availability but it should not be calculated in such oversimplified way. 6 Line 327, 332: “manuscript” should be paper or study.

Round 2

Reviewer 2 Report

My concerns arising in the previous review have been well addressed accordingly and the quality of the manuscript is much improved. The authors have added the delineation method description in Section 2.4 with a supplementary figure. The water balance equation in Section 2.2 has been clearly presented and the quality of figures is now more legible with a high resolution. The unrelated parts existing in the previous version such as the general description of the GIS approach for watershed delineation has been removed. The present form of manuscript is now more focused. Overall, the method description is clearer and the conclusions are well summarized. Given the scientific value of its nature to local development, I recommend to accept for publication after further improvement of English writing and correction of any typos.